# Extracellular Matrix Components as Diagnostic Tools in Inflammatory Bowel Disease

**DOI:** 10.3390/biology10101024

**Published:** 2021-10-11

**Authors:** Laura Golusda, Anja A. Kühl, Britta Siegmund, Daniela Paclik

**Affiliations:** 1Freie Universität Berlin and Humboldt Universität zu Berlin, iPATH.Berlin, Campus Benjamin Franklin, Charité—Universitätsmedizin Berlin, 12200 Berlin, Germany; laura.golusda@charite.de (L.G.); anja.kuehl@charite.de (A.A.K.); 2Freie Universität Berlin and Humboldt Universität zu Berlin, Medizinische Klinik für Gastroenterologie, Infektiologie und Rheumatologie (einschl. Arbeitsbereich Ernährungsmedizin), Campus Benjamin Franklin, Charité—Universitätsmedizin Berlin, 12203 Berlin, Germany; britta.siegmund@charite.de

**Keywords:** extracellular matrix, glycosaminoglycans, inflammatory bowel disease, ulcerative colitis, Crohn’s disease, fibrosis, stenosis, magnetic resonance imaging, elastography, histopathology

## Abstract

**Simple Summary:**

For decades, the extracellular matrix (ECM) has been defined as a structure component playing a rather neglected role in the human body. In recent years, research has shed light on the role of ECM within cellular processes, including proliferation, migration and differentiation, as well as in inflammation. In inflammation, ECM composition is constantly being remodeled and undergoes dynamic and rapid changes. Tracking these changes could serve as a novel diagnostic tool. Inflammatory bowel disease is accompanied by complications such as fibrosis, stenosis and fistulas. All of these structural complications involve excessive synthesis or degradation of ECM. With this review, we explored whether the analysis of ECM composition can be of support in diagnosing inflammatory bowel disease and whether changes within ECM can help to predict a complicated disease course early on.

**Abstract:**

Work from the last years indicates that the extracellular matrix (ECM) plays a direct role in various cellular processes, including proliferation, migration and differentiation. Besides homeostatic processes, its regulatory function in inflammation becomes more and more evident. In inflammation, such as inflammatory bowel disease, the ECM composition is constantly remodeled, and this can result in a structuring of fistulizing disease course. Thus, tracking early ECM changes might bear the potential to predict the disease course. In this review, we provide an overview of relevant diagnostic methods, focusing on ECM changes.

## 1. Introduction

The main forms of inflammatory bowel disease (IBD) are ulcerative colitis (UC) and Crohn’s disease (CD). Both are chronic inflammatory conditions with an altered extracellular matrix (ECM). The diagnosis of UC and CD is a lifelong threat, as the available therapies treat and ease the symptoms but do not cure the disease. It is accepted that IBD results from an exaggerated mucosal immune response in genetically predisposed individuals. Environmental factors trigger this response, and a leaky epithelial barrier is either a cause or consequence. The onset of IBD occurs in late adolescence and early adulthood affecting all aspects of life. The incidence and prevalence of IBD is increasing worldwide just as the total number of related deaths [1]. The Western Europe region had the highest age-standardized death rate in 2017 [1]. Overall, an estimated 1.3 million people in Europe suffer from IBD, which equals 0.2% of the European population [2]. The amount of direct healthcare costs per patient per year reach up to €2000 (UC) and €3500 (CD), respectively [2]. It is still not clear whether changes in ECM occur at an early disease stage triggering inflammation and contributing to chronicity or changes develop at a later disease course that are caused by chronic and excessive inflammation. Complicated disease (stricturing or penetrating disease behavior) is a consequence of altered ECM requiring intervention, such as balloon dilation or surgery, including strictureplasty or resection [3]. Over 50% of CD patients and up to 11% of UC patients experience fibrostenotic complications [4]. These complications are accompanied by changes in the ECM. Currently, the diagnosis of IBD is based on a multitude of parameters from clinics, laboratory, imaging, endoscopy and histopathology [5,6]. However, the currently available tools to predict disease course have not entered clinical routine yet [7]. Thus, an in-depth analysis of ECM over the course of the disease might provide a novel tool to fill in this gap.

The core components of the ECM are fibronectin, collagens, laminins and proteoglycans. Proteoglycans have a protein core to which sulfated glycosaminoglycans (s-GAGs) are attached. The attached s-GAGs are linear polysaccharides, which are highly negatively charged. They are build out of disaccharide building blocks. Based on the degree of sulfation, the position of sulfation, the linkage between each and the type of monomeric unit, they are classified into the following groups: chondroitin sulfate/dermatan sulfate (CS/DS), keratan sulfate (KS) and heparin/heparan sulfate (HS) [8,9]. Hyaluronic acid is the only non-sulfated GAG present in the ECM. GAGs take part in cell–matrix interactions, and s-GAGs are strongly expressed in the ECM of intestinal tissues. Throughout the gastrointestinal tract, s-GAGs are found in the subepithelial basal membrane, the vascular endothelium and the ECM of the (sub)mucosa [10]. Naba et al. characterized the matrisome that defines all colon-tissue-specific proteins in the mouse colon that are part of or associated with the ECM [11,12].

This review presents the latest findings on ECM changes in IBD, and by this, it illustrates how these could not only serve as a tool to monitor but also to predict the disease course. Figure 1 illustrates the ECM changes involved in fistulizing and sttricturing disease.

## 2. Materials and Methods

A search of the literature in MEDLINE, using the electronic database PubMed, was conducted to identify GAG/ECM-associated diagnostic tools in IBD. The following search terms were included: ((inflammatory bowel disease) OR (ulcerative colitis) OR (Crohn’s disease)) AND (diagnosis) AND ((magnetic resonance imaging) OR (sonography) OR (ultrasound) OR (elastography) OR (histology) OR (histopathology)) AND ((extracellular matrix) OR (glycosaminoglycan)) AND (fibrosis OR stenosis OR stricture OR fistula), as well as (contrast-enhanced mri) AND (bowel) OR (intestine). Included were studies in adult humans of the past 10 years.

## 3. Results

### 3.1. In Vivo Imaging

Several types of imaging techniques are required for diagnosing UC and CD (Table 1). This generally includes colonoscopy and, initially, small-bowel magnetic resonance imaging (MRI), as well as upper endoscopy, to determine the disease pattern and the type of disease in the individual patient. Computed tomography (CT) should be restricted to emergency situations in order to limit radiation exposure. Furthermore, ultrasound (US) is nowadays frequently used for follow-up evaluation [13,14]. Employing CT enterography in order to predict small-intestine fibrosis in CD patients revealed that the only predictive parameter was mesenteric hypervascularity; however, this parameter correlates better with inflammation than fibrosis [15]. Combining CT enterography with positron emission tomography [PET] (PET/CT) was as accurate as PET/magnetic resonance enterography in detecting strictures and bowel wall thickening [16]. However, for both techniques, the accuracy depended on the bowel segment. The accuracy was low for duodenum and colon, and it was highest for the terminal ileum and cecum, as well as ileocolonic anastomosis [16]. For the detection of fistulas, the sensitivity, specificity and accuracy of small-intestine contrast ultrasonography (SICUS) are comparable to CT enteroclysis and surgical findings [17]. SICUS is also accurate in detecting strictures and their extension in the proximal and distal small intestine, as well as fistulas and abscesses, when compared to surgical and histological findings [18]. Compared to intraoperative findings, SICUS is more accurate in assessing fistulas, abscesses and strictures than dilatations and bowel-wall thickening [19]. The treatment response was monitored in a multicenter study including patients with active CD by US assessing bowel wall thickness, vascularization and contrast-agent uptake, as well as fistulas, abscesses and stenoses [20]. For determining CD activity, Novak et al. developed and validated a US score (simple sonographic score) [21]. This score might have been too simple and was not widely accepted. Besides this, the Limberg score is widely used to assess disease activity with US [22]. A recent study proposed a regression model based on multimodal multi-parametric ultrasound to assess CD activity [23]. In order to differentiate between inflammation and fibrosis, Bhatnagar et al. compared sonographic features with histomorphology in CD patients [24]. Their study revealed that thickness of the mucosal layer, rather than bowel-wall thickness, correlates with acute inflammation, chronic inflammation and fibrosis [24]. Fibrosis was associated with reduced submucosal echogenicity, increased submucosal echogenicity with hypoechoic bands and an ill-defined submucosa [24].

The first contrast-enhanced MRI was performed in 1981 [25]. Oral ferric chloride and inhaled oxygen was used. Resulting in altered spin-lattice relaxation time of the fundus and was suggested to be useful as a bowel-labeling agent [25]. Since 1988, the paramagnetic contrast agent gadolinium diethylene triamine penta-acetic acid (Gd-DTPA) has been in clinical application in Germany, the USA and Japan [26]. Intravenous Gd-DPTA administration improved MRI with regard to detecting disease complications and extent of bowel involvement [27]. Distribution of i.v.-injected Gd-based contrast agents is associated with blood supply, but there is no tissue targeting or specificity [28]. Contrast agents targeting extracellular matrix for detection of aberrant matrix formation are still in the preclinical phase. Stable plaque formation in atherosclerosis is associated with ECM accumulation [29], and it has been visualized by very small superparamagnetic nanoparticles [30]. ECM components, such as collagen or fibrin–fibronectin complexes, were targeted in models of liver fibrosis and colorectal cancer [31,32]. To our knowledge, there are no data available for ECM-targeting contrast agents in models of intestinal fibrosis. A prospective multicenter study compared the diagnostic accuracy of MR enterography and US in small-bowel CD and concluded MR enterography more sensitive and more specific [33]. Compared to SICUS, MR enterography is more accurate in assessing fistulas and strictures, while it is comparably accurate in assessing abscesses and dilatations [19]. Additionally, contrast-enhanced MR enterography is preferred to biochemical markers, as a significant number of patients with quiescent disease have high fecal calprotectin levels [34]. Furthermore, MR imaging could accurately detect and distinguish varying degrees of bowel fibrosis with or without coexisting inflammation. Magnetization transfer MRI could accurately detect the severity of bowel fibrosis in stricturing CD but not the inflammatory component within the stricture [35], whereas fibrotic and inflammatory strictures could be differentiated from purely inflammatory strictures [35]. With reasonable accuracy, an area under the ROC curve of > 0.7 MRI could distinguish inflammation/edema and muscular hypertrophy from fibrosis in ileal CD in a retrospective study [36].

**Table 1 biology-10-01024-t001:** Diagnostic imaging methods to detect inflammation and complications in IBD.

Method	Diagnosis	References
Ultrasound	Bowel wall thickening, lesions, hypervascularity, strictures, dilatation	[37,38,39]
Computed tomography	Mesenteric hypervascularity, fistulae, abscesses	[15,37]
PET/CT	Strictures, bowel wall thickening, inflammatory activity	[16,40]
Magnetic resonance imaging	Strictures, fistulae, abscesses, dilatations, edema, muscular hypertrophy	[19,36]
SICUS	Strictures, fistulae, abscesses	[18,19]

Ten years ago, a small number of CD patients underwent ultrasound elastography (US-E) prior to elective resection of small-intestine strictures in a pilot study. US-E was guided by MRI and CT, and the scanning included diseased stenotic and adjacent unaffected small intestine. This pilot study revealed that lower strain values indicate stiffer tissue in stenotic bowel compared to unaffected bowel [41]. Histopathology confirmed predominantly fibrotic strictures with submucosal collagen depositions, while stenotic tissue was characterized by fibrosis and mild-to-moderate inflammation [41]. US-E is a feasible and reproducible technique for assessing ileal wall fibrosis in CD patients, as the strain ratio correlates significantly with the severity of fibrosis [42]. US-E detects fibrosis in ileal/ileocolonic segments of CD patients via increased muscular layer thickness and collagen deposition [43], but it is still not able to differentiate fibrosis from inflammation even when contrast enhanced [44,45]. Shear-wave elastography (SWE) is a novel technique that allows for quantitative estimation of tissue stiffness. While resected bowel segments showed ex vivo a higher mean shear-wave speed in high-grade fibrosis than low-grade fibrosis with minimal overlap [46], there was no relationship of fibrosis and SWE in vivo [47]. Lu et al. found that fibrosis was not the major component in bowel-wall thickening and strictured bowel segments, but rather muscular hypertrophy [47]. When combining SWE and the Limberg score for bowel vascularization, high- and low-grade inflammation, as well as high- and low-grade fibrosis, could be discriminated [48]. SWE seems also feasible for determining disease activity in UC, while it is rather discriminative in the left-sided than in the right-sided colon [49].

Frequent clinical applications of magnetic resonance elastography (MRE) in the abdomen are made for liver diseases, whereas the bowel is technically challenging, owing to its location, mobility and physiological motion [50]. These challenges were tackled in a prospective pilot study. Here, shear-wave speed and loss angle, representing stiffness and solid–fluid behavior, were studied in IBD patients and healthy controls [51]. Both were increased in IBD patients compared to controls; however, there was no significant difference between UC and CD [51]. Further studies are needed for an assessment of intestinal fibrosis.

### 3.2. Histopathology

In UC, histopathology is required for diagnosis, assessment of disease activity and identification of dysplasia and cancer [14]. Microscopic features include mucosal architecture, lamina propria cellularity, neutrophil granulocyte infiltration and epithelial abnormality [14] (Table 2). Microscopic features in diagnosing CD are discontinuous chronic inflammation and crypt distortion, as well as granulomas [13]. In UC, any colonic stricture is suspicious of cancer and requires evaluations as such [14]. In contrast, structuring disease in CD can occur even in the absence of cancer. Distinguishing structuring disease from cancer can be a diagnostic challenge [13]. However, an international consortium developed and validated a stricture histopathology scoring system, in order to enable the development of novel biomarkers and support the construction of imaging endpoints for clinical trials in stricturing CD [52]. The consensus was reached for evaluating tissue sections stained with hematoxylin and eosin (H&E), but not using Movat or elastin stains. No consensus could be reached for the need of trichrome stain [52]. The Movat stain is a pentachrome stain and differentiates elastic, collagen and reticular fibers, as well as muscle and fibrin [53], whereas a trichrome staining allows for differentiation of muscle fibers, collagen and keratin [54]. A modification of Movat pentachrome stain was developed for the demonstration of elastin, sulfated macromolecules (proteoglycans, glycosaminoglycans and secreted mucins), collagens, myofibrils and erythrocytes [55]. Hence, ECM components or glycosaminoglycans are not yet part of the clinical routine for diagnosis or follow-up but are potential candidates in histopathological assessment. For example, serum levels of matrix metalloproteinase (MMP)-9 have been shown to be increased in UC patients. In line, within the mucosa immunohistochemically detected MMP-9 expression increases with severity [56]. The advantage of histopathology is the spatial allocation of protein expression. For instance, Fonseca-Camarillo et al. identified upregulation of extracellular matrix metalloproteinase inducer (EMMPRIN, CD147) in patients with active UC and mononuclear and endothelial cells being the main producers [57]. The numbers of CD147-positive cells expressing MMP-23 and MMP-10 are increased in active UC compared to active CD [57]. Analyzing stenotic and inflamed ileum from CD patients revealed enhanced CD3-positivity in the inflamed region and increased collagen-positivity in the stenotic region [58]. The increased collagen content was accompanied by increased losyl oxidase (LOX) involved in the process of collagen deposition linking collagen to elastin but also interacting with fibronectin [58]. Moreover, CD20-positive cells were increased in fistulizing versus stenosing CD [42]. The ECM glycoprotein tenascin C was also upregulated in the mucosa of patients with UC and CD compared to controls [59]. Tenasin C is mainly expressed in the lamina propria [59]. Additionally, mucosal mRNA expression has been associated with treatment response, and tenascin C mRNA expression was higher in UC patients nonresponsive to infliximab therapy [59].

### 3.3. Serological Markers

Intestinal mesenchymal cells (MSCs), including fibroblasts, myofibroblasts and smooth muscle cells, are the main secretors of ECM [60]. MSCs are non-epithelial, non-endothelial and non-hematopoietic cells [61]. In the intestine, various MSC populations occupy distinct niches and perform site-specific functions [61]. Increased MSC numbers and excessive ECM secretion are hallmark features of intestinal strictures. Myofibroblasts, fibroblasts and smooth muscle cells differ in their expression of vimentin, α-smooth muscle actin and desmin [61,62]. During intestinal inflammation, MSCs differentiate and de-differentiate between these three phenotypes [63]. While fibrosis develops to preserve tissue architecture and functions as an integral part of wound healing and tissue repair [64], pro-inflammatory and pro-fibrotic mediators in IBD constantly activate myofibroblasts, leading to an ECM overproduction and fibrosis [65] resulting in stricturing disease [4,60]. Moreover, pericytes surrounding blood vessels express typical fibroblast markers under inflammatory conditions and produce large quantities of ECM components [66]. Furthermore, epithelial-mesenchymal transition and endothelial-mesenchymal transition play a role in intestinal fibrosis in IBD [67,68], that need further elucidation. As the balance of degradation and production of ECM is disrupted, components of the degraded ECM can be found in the peripheral blood. Therefore, assessment of degraded ECM components in the serum has diagnostic potential (Table 3). Assessing s-GAGs, hyaluronan and soluble CD138 in the serum revealed no changes in CD patients compared to healthy controls, whereas hyaluronan was significantly increased in UC patients and correlated with Mayo score and thus disease severity [69]. Hence, this difference might help to distinguish UC from CD. However, the data are currently limited to treatment-naïve patients. Steroid therapy resulted in an increase of hyaluronan and was statistically not significantly different from untreated UC or adalimumab-treated UC patients [69]. Yamaguchi et al. found a correlation between the serum-derived hyaluronan-associated protein (SHAP) and disease activity in UC and CD [70] questionable. Specific ECM degradation proteins with diagnostic value, such as MMP-9 degraded type III collagen fragment C3M, have been associated with penetrating CD [71]. Serum levels of C3M were increased in patients with penetrating CD when compared to healthy controls but neither to non-penetrating or stricturing disease nor to perianal fistula [71]. Mortensen et al. also defined biomarker combinations to discriminate CD from UC and UC from non-IBD controls [72]. Serum levels of VICM (MMP-2/8 degraded and citrullinated-vimentin), C3M and C4M (MMP-9 degraded collagen type IV) discriminate CD from UC, and C1M (MMP-9 degraded collagen type I) and C3M discriminate UC from non-IBD [72]. Compared to healthy controls, the tissue inhibitor of metalloproteinase 1 TIMP-1 was increased in the serum of patients with UC or CD and higher in active disease, allowing also for disease activity assessment [73]. Using serum glycoproteome profiles, Stidham et al. identified two biomarkers which distinguish inflammatory from fibrostenotic phenotypes of CD [74]. Both cartilage oligomeric matrix protein (COMP) and hepatocyte growth factor activator (HGFA) showed ≥ 20% change in relative abundance between fibrotic and inflammatory disease types [74]. Van Haaften et al. identified that serum levels of formation and degradation products of collagens can serve to differentiate penetrating and non-stricturing/non-penetrating, as well as stricturing CD in the terminal ileum [75]. Other studies comparing different ECM components found that a strong increase of extracellular matrix protein 1 (ECM1) in CD patients is correlated with a higher risk to change from inflammatory phenotype to stricturing phenotype [76]. Unfortunately, for none of the markers exists a standard value which clearly defines the disease status or activity.

While it is not known if the break of the intestinal barrier is a cause or consequence of IBD, it is a feature of UC and CD, and microbial products also influence the ECM [77]. Antimicrobial antibodies against specific bacteria, bacterial membrane components or glycans, such as anti–*Saccharomyces cerevisiae*, anti–*Escherichia coli* outer-membrane porine C, anti-flagellin, anti-laminaribioside carbohydrate antibodies, anti-mannobioside carbohydrate antibodies, anti-chitobioside carbohydrate antibodies, anti-chitin antibody and anti-laminarin antibodies, can be used as serum markers [78,79,80]. Amongst anti-glycans, no correlation with fibrostenotic stricture could be shown, while serum levels of anti-zymogen granule glycoprotein 2 may aid as a tool for diagnosis and differentiation of CD and could indicate a more complicated CD course and anti–*Saccharomyces cerevisiae* antibodies (ASCA) were qualitatively and quantitatively linked to CD, CD complications and need for surgery [81]. A meta-analysis even indicated that positive ASCA status is a risk factor for early onset age, ileal involvement, complicated behavior, perianal disease and requirement for surgery in CD [82]. Additionally, anti–*Escherichia coli* outer-membrane porin C is associated with Crohn’s disease phenotypes, and patients with the highest level of serum reactivity toward an increasing number of microbiota have the greatest frequency of strictures, internal perforations and small-bowel surgery [79]. Papadakis et al. found an association of anti-flagellin with fibrostenosis, penetrating disease and small-bowel involvement, as well as surgery [83]. There is no reported association of fibrostenotic Crohn’s disease and antibodies directed against laminaribioside carbohydrate antibodies, antichitobioside carbohydrate antibodies or mannobioside carbohydrate [84,85,86]. Whereas anti-laminarin and anti-chitin correlate could be important serologic markers for the prediction of CD-related complications and surgery [87]. Overall, higher titers and several seroreactivities pose an increased risk to develop a complicated disease course [84].

Additionally, some growth factors known to influence tissue repair, collagen secretion, angiogenesis or fibroblast proliferation have been studied as possible biomarkers to detect early ECM alterations and fibrosis. Among them are vascular endothelial growth factor (VEGF), platelet-derived growth factor (PDGF), fibroblast growth factor (bFGF) and human-chitinase-3-like 1, influencing angiogenesis, fibrogenesis, myofibroblast proliferation and myofibroblast-induced collagen secretion [88]. VEGF and bFGF both promoting angiogenesis, tissue repair and fibroblast proliferation are significantly increased in serum of CD patients and correlate with bowel wall thickness [89]. The expression of PDGF is enhanced at sites of inflammation and fibrosis [90], as well as in the serum of CD patients [91]. None of the markers can be used to predict fibrostenotic risk, but they are markers for the severity of fibrotic changes.

Besides components of ECM, growth factors and bacterial components, a potential tool to diagnose fibrosis is the detection of epigenetic markers, such as microRNA (miR). These short non-coding RNAs regulate the expression of target genes at a post-transcriptional level. One of those, miR-200b, was shown to be increased in liver fibrosis [92]. When comparing CD patients with and without fibrostenotic complications, there is a significant difference in serum levels of miR-200b between the groups [92]. Mehta et al. revealed that a downregulation of miR-200b in intestinal epithelial cells is associated with epithelial to mesenchymal transition [93]. Furthermore, low serum levels of miR-19 [94] and miR-29b [95] could be correlated with a stricturing phenotype in CD patients. Currently there are no epigenetic biomarkers which allow for the early prediction of a high risk to develop fibrostenotic complications, and further studies to characterize their role are essential. To summarize this paragraph, a number of potential biomarkers have been described to identify fibrostenotic complications. Still, due to a low sensitivity and specificity, none of them has entered clinical routine.

**Table 3 biology-10-01024-t003:** Serological marker to discriminate intestinal inflammation from healthy gut.

Serological Marker Group	Detected Component	References
ECM components	Hyaluronan, C3M, C4M, VICM, COMP, TIMP-1, ECM1	[69,71,72,73,74,76]
Microbial substances	Bacteria, bacterial membrane components	[78,79,80]
Growth factors	VEGF, PDGF, bFGF, human-chitinase-3-like 1, HGFA	[74,88,89,90]
microRNA	miR-200b, miR-19, miR29b	[92,93,94,95]

## 4. Discussion

Changes in the ECM trigger inflammation and contribute to chronicity in IBD. This is reflected by the presence of fibrosis and stricturing disease in about 11% of UC patients, as well as over 50% of CD patients. These complications often require surgical intervention. We here review the latest developments in diagnosing ECM changes in order to assess complicated disease, but also to monitor mucosal healing or differentiate CD from UC.

Early and accurate diagnosis of a complicated disease course is crucial for assessment and management of these patients. For example, intestinal fibrosis negatively influences the response to therapy with biologicals [96,97].

The gold standard for diagnosing disease remains endoscopy. Cross-sectional imaging techniques add, in particular, information with regards to the small bowel, as well as to complications, including abscess and fistulizing disease. There is European consent that the use of CT should be limited to emergency due to radiation exposure. Thus, MRI should present the standard technique and has a high contrast resolution, providing anatomical details without ionizing radiation, but is time- and cost-intensive. Additionally, most contrast agents used in MRI contain gadolinium, which can accumulate in tissues, regardless of renal function [98]. Against a background of IBD patients having an increased risk to develop chronic kidney disease [99] and of patients with impaired renal function developing in rare cases gadolinium-associated systemic fibrosis [100], radiologist have to balance risks and benefits of gadolinium-enhanced MRI.

The technique of US is rapid, safe and easy to use. Recent studies have provided convincing evidence that US can be performed in a reproducible manner; thus, the former argument that it strongly dependent on the examiner is outdated [101]. Elastography is also non-invasive, and the advantages of US apply to elastography. Shear-wave elastography requires fasting of the patients in order to reduce bowel content and blood flow. The US-E strain ratio not only depends on the degree of pressure exerted by the US probe [42], but the mesenteric tissue surrounding the bowel wall serves as control. This control might be misleading, as hyperplasia of mesenteric fat itself already affects the strain ratio. Hyperplastic mesenteric fat wrapping around the circumference of the intestine (creeping fat) is a common feature in CD [102,103]. Additionally, mesenteric and creeping fat are inflamed in CD [104], providing misleading strain ratios [42]. The application field of elastography is limited to selected bowel segments and allows no cross-sectional imaging [105]. Multimodal imaging would be optimal for assessing the disease, the disease activity and complications, but one has to keep in mind that bowel peristaltic negatively influences all imaging techniques. There is no reference standard in the diagnosis of IBD, but as an initial diagnostic tool, radiologic visualization, combined with a follow-up via US, is widely used to diagnose and evaluate IBD [5].

In situ imaging using histopathology provides a clear picture of the intestinal tissue, but it is only a snapshot and is limited to the surface layers when taken as biopsy. Histopathology, in combination with endoscopy, gives a good overview of stricturing, mucosal surface and gut motion and is one of the most important diagnostic strategies. One disadvantage is that not all segments can be reached, and the view is restricted to the luminal surface. In this regard, adding insult to injury during diagnosis, noninvasive diagnostic tools are favorable.

Using serological biomarkers presents an especially minimal invasive and fast approach. Various ECM-related biomarkers not only diagnose IBD but also differentiate CD from UC. However, due to a low sensitivity and specificity, none of them has entered clinical routine. Besides this, serum biomarkers bear the risk of capturing ECM changes in other organs, since IBD is associated with extraintestinal manifestations, such as rheumatological, musculoskeletal, hepatological and dermatological manifestations; arthropathies and uveitis are also frequent [106]. Nevertheless, additional assessment of ECM changes provides a great potential tool in IBD diagnosis.

## 5. Conclusions

Various methods and techniques are available for the diagnosis of UC and CD, as well as the assessment of disease activity and complications. Every technique has its advantages and accuracy but also implies disadvantages and inaccuracies. There is no one-size-fits-all. The optimal treatment of IBD patients should aim at a multimodal approach. Some techniques and approaches have not made it into the clinic yet and need further development and validation. Involving the extracellular matrix and its synthesis, changes and degradation provide a potential toolbox to monitor disease course and phenotype over time.

## Figures and Tables

**Figure 1 biology-10-01024-f001:**
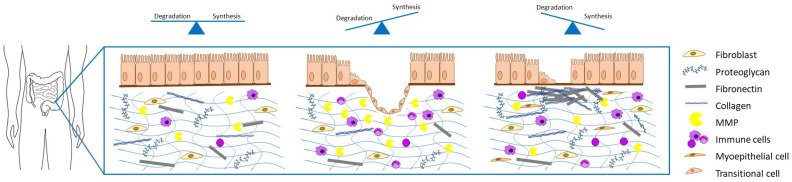
Changes in ECM composition during a chronic disease course. The panels shown illustrate homeostasis (**left**), fistulizing disease (**middle**) and stricturing disease (**right**).

**Table 2 biology-10-01024-t002:** Diagnostic histopathological methods to detect inflammation and complications in IBD.

Histological Stainings	Detected Features	References
H&E staining	Mucosal architecture, lamina propria cellularity, neutrophil infiltration, epithelial abnormality, granuloma	[13,14]
Trichrome staining	Muscle fibers, collagen, keratin	[54]
Pentachrome staining	Elastin, collagen, reticular fibers, muscle, fibrin	[53]
Immunohistochemical antibody staining	Detection of immune cell composition, matrix metalloproteinases, matrix metalloproteinase inducers	[42,56,57,58]

## Data Availability

Not applicable.

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
