# Peer review of "Extracellular Matrix Components as Diagnostic Tools in Inflammatory Bowel Disease"

_biology, 2021, doi:10.3390/biology10101024_

Round 1
Reviewer 1 Report
In the present paper the authors aimed to provide the reader with an accurate overview of the existing diagnostic tools used in inflammatory bowel disease (IBD), focusing on extracellular matrix (ECM) changes. The study is very well structured, presenting numerous diagnostic methods, amongst those being imaging tools, histopathology and serological markers. Overall, the paper is interesting, it contains up to date information and it certainly is clinically relevant.
However, some issues should be addressed, such as:
- The paragraph describing the history of MRI (lines 119-125) should be shorter in order to not deviate from the purpose of the paper.
- The markers presented in subchapters 3.2 and 3.3 should also be structured in tables as it was done in subchapter 3.1 for a better comprehension of the information provided.
- Citations 53 and 88 that were co-authored by some of the authors of this paper should be replaced if possible.
Author Response
We thank the reviewers for their useful comments. All their remarks to improve the manuscript have been addressed in the revised version of the manuscript.
A point to point response to the comments raised by the reviewers follows below:
Reviewer #1
In the present paper the authors aimed to provide the reader with an accurate overview of the existing diagnostic tools used in inflammatory bowel disease (IBD), focusing on extracellular matrix (ECM) changes. The study is very well structured, presenting numerous diagnostic methods, amongst those being imaging tools, histopathology and serological markers. Overall, the paper is interesting, it contains up to date information and it certainly is clinically relevant.
However, some issues should be addressed, such as:
- The paragraph describing the history of MRI (lines 119-125) should be shorter in order to not deviate from the purpose of the paper.
We thank the reviewer for this suggestion and have shortened the respective paragraph.
- The markers presented in subchapters 3.2 and 3.3 should also be structured in tables as it was done in subchapter 3.1 for a better comprehension of the information provided.
We thankfully addressed this point and included tables for the subchapters 3.2 and 3.3.
- Citations 53 and 88 that were co-authored by some of the authors of this paper should be replaced if possible.
We agree with the reviewer, that self-citations should be avoided. To our knowledge, the manuscript of Reiter et al. describes for the first time a pilot study of MRE in patients with IBD.
A PubMed search with the keywords Crohn AND “inflamed creeping fat” resulted in 11 papers with 6 being authored by Britta Siegmund, who has a long-standing experience in IBD and creeping fat. The paper by Kredel LI et al. not only deals with creeping fat and inflammation in Crohn´s disease but also with the immune cell composition within the creeping fat. Based on this literature search, we decided to keep these citations.
Reviewer 2 Report
In this Review, the authors analyze and discuss the possible role of changes in the intestinal extracellular matrix (ECM) occurring in inflammatory bowel disease (IBD) as a novel diagnostic tool. In particular, whether the analysis of ECM composition can be of support in diagnosing IBD and whether changes within ECM can help to predict the clinical course of the disease.
The paper is interesting and well written, but there are some points that the authors should take into consideration and better clarify, namely:
- Section 3.1. In-vivo imaging is mostly a general description of the performance of various imaging techniques (US, CT, PET / CT, MRI, US-E) and less related to the specific ECM alterations that develop in IBD.
- Section 3.2. Histopathology is too generic and succinct, while it should be integrated with further data regarding the specific ECM abnormalities reported in IBD.
- The paragraph on anti-microbial antibodies (lines 244-250) is presented and described in a manner completely unrelated to ECM changes in IBD. Contrary to what has been reported, the increase in some of the anti-microbial antibodies is associated with fibrostenotic Crohn's disease.
- The paragraph on growth factors (lines 251-260) is entirely generic and unrelated to the specific alterations of the ECM in IBD. Same consideration for the next paragraph on miRs (lines 261-274).
Author Response
We thank the reviewers for their useful comments. All their remarks to improve the manuscript have been addressed in the revised version of the manuscript.
A point to point response to the comments raised by the reviewers follows below:
Reviewer #2
- Authors might discuss extensive literature on the role of hyaluronan and associated molecules that have been identified as a major ECM component altered in this disease.
We totally agree with the reviewer about the role of hyaluronan as a non-sulfated GAG being a major component of the ECM. Additionally, the content of hyaluronan is altered during IBD and hyaluronan shows behavior of a viscoelastic solid based on the Kelvin-Voigt model. Hence, elasticity should be determined by the proportion of hyaluronan. Unfortunately, studies focusing on hyaluronan as a diagnostic target in fibrostenotic IBD are scarce. We included the study of Yamaguchi Y et al., who found a correlation between the serum-derived hyaluronan-associated protein (SHAP) and disease activity in UC and CD (line 242 ff). Decades ago, studies on luminal hyaluronan in CD patients showed an increase of hyaluronan in perfusion fluids of patients with active disease compared to patients in remission. This finding never approached clinical relevance and will not be mentioned in the review [Colombel JF et al., 1989; Ahrenstedt O et al., 1992].
- The review would benefit by some discussion of postulated pathways (cellular and non cellular) leading to the pathology that results from this disease.
This is a very good suggestion, thus, this topic has been included in the section about the cellular sources of ECM in order to keep the structure of the review and stay in its boundaries.
- Some discussion of the cellular sources of theses ECM changes might add interest to the review.
We thank the reviewer for this suggestion and have now added some information about ECM-secreting cells (line 219 ff).
- Movats stain is also useful to distinguish accumulation of proteoglycans and hyaluronan
This is now added to the manuscript (line 193 ff).
Reviewer 3 Report
Authors might discuss extensive literature on the role of hyaluronan and associated molecules that have been identified as a major ECM component altered in this disease.
The review would benefit by some discussion of postulated pathways (cellular and non cellular) leading to the pathology that results from this disease.
Some discussion of the cellular sources of theses ECM changes might add interest to the review.
Movats stain is also useful to distinguish accumulation of proteoglycans and hyaluronan
Author Response
We thank the reviewers for their useful comments. All their remarks to improve the manuscript have been addressed in the revised version of the manuscript.
A point to point response to the comments raised by the reviewers follows below:
Reviewer #3
In this Review, the authors analyze and discuss the possible role of changes in the intestinal extracellular matrix (ECM) occurring in inflammatory bowel disease (IBD) as a novel diagnostic tool. In particular, whether the analysis of ECM composition can be of support in diagnosing IBD and whether changes within ECM can help to predict the clinical course of the disease.
The paper is interesting and well written, but there are some points that the authors should take into consideration and better clarify, namely:
- Section 3.1. In-vivo imaging is mostly a general description of the performance of various imaging techniques (US, CT, PET / CT, MRI, US-E) and less related to the specific ECM alterations that develop in IBD.
We thank the reviewer for this comment and clarified this point in the introduction.
This review concentrates on three diagnostic tools for assessing and monitoring ECM changes, the in-vivo imaging, the histopathology and the serology. All these techniques are routinely used in the clinic.
- Section 3.2. Histopathology is too generic and succinct, while it should be integrated with further data regarding the specific ECM abnormalities reported in IBD.
We agree, that the term histopathology is a generic term. It subsumed all histopathological techniques including histochemistry, immunohistochemistry, immunofluorescence, molecular pathology, and in-situ hybridization. This review focusses on studies elucidating the role of an altered ECM as a diagnostic tool. Unfortunately, the whole potential of histopathology within this field has not been exploited. This is best reflected in the study of Gordon et al. [2021 Gut], where a consensus could be reached for H&E-stained sections for evaluation but not for pentachrome, which would add so much more information.
- The paragraph on anti-microbial antibodies (lines 244-250) is presented and described in a manner completely unrelated to ECM changes in IBD. Contrary to what has been reported, the increase in some of the anti-microbial antibodies is associated with fibrostenotic Crohn's disease.
We wanted to add another layer to intestinal ECM and serum markers as the microbiota plays a role in fibrogenesis (e.g. by enhancing TGFβ production by epithelial cells). Now, an introductory sentence is added.
We thank the reviewer for pointing out a more thorough literature research. We corrected the point, that anti-glycans show no correlation with fibrostenotic stricture formation and added further references e.g., Degenhardt F et al., 2006 (line 273 ff).
- The paragraph on growth factors (lines 251-260) is entirely generic and unrelated to the specific alterations of the ECM in IBD. Same consideration for the next paragraph on miRs (lines 261-274).
Currently, there are no biomarkers which reliably predict the risk of developing intestinal strictures or identify early stages of fibrosis prior to clinical symptoms. Candidate biomarkers of intestinal fibrosis include gene variants e.g., NOD2 or serum markers like microRNAs, ECM proteins or enzymes, growth factors anti-microbial antibodies and circulating cells. Our review is intended to give an overview of potential candidates for diagnosing and/or predicting complicated IBD focusing on ECM components or mediators of ECM accumulation/degradation. We do not want to put emphasis on these markers, but we think that they should be mentioned and discussed.